# A Question-Based Method to Calculate the Human Appropriation of Land for Food (HALF) Index

**Marije Hoff** [1,2] **and Hugo Jan de Boer** [1,*]

1   Environmental Sciences, Faculty of Geosciences, Copernicus Institute of Sustainable Development, Utrecht University, Princetonlaan 8a, 3584 CB Utrecht, The Netherlands; marije.c.hoff@gmail.com
2   Department of Science, University College Utrecht, P.O. Box 80145, 3508 TC Utrecht, The Netherlands
*   Correspondence: h.j.deboer@uu.nl

**Abstract:** Global dietary consumption strongly determines agricultural land requirements. Yet, it is currently difficult for individual consumers to quantify the environmental impact of their individual diet. One relatively easy to understand metric is the Human Appropriation of Land for Food (HALF) index. The HALF index expresses the global land area percentage required for food production were the global population to consume one specific diet. Calculation of the HALF index is not trivial, making the index not suitable for individual consumers to assess their personal diet. The aim of this research is to develop and test a new method to calculate a personalized HALF index based on a limited set of multiple-choice questions that can be answered by a typical consumer. Considering the sensitivity of the original HALF index, we developed a set of ten multiple-choice questions that focus on the type and quantity of consumed animal products in addition to staple foods and overall consumption quantity. To illustrate a potential implementation, we present our question-based HALF index calculator in the form of an online graphical user interface. Across a sample of 23 country-specific diets, the question-based HALF index closely matches the original HALF index with a regression slope of near unity ($r^2 = 0.94$, $p < 0.001$). Our results indicate that the question-based HALF index can be used by individual consumers to quantify the consequences of their dietary choices on land use for agriculture.

**Keywords:** HALF; diet; changing consumption patterns; land requirements

## 1. Introduction

Currently, 37% of the terrestrial land surface is used for agricultural food production [1]. Agricultural land use greatly impacts the global water, nutrient, and carbon cycles and the conversion of (semi)-natural lands to agriculture leads to loss of biodiversity [2–5]. Furthermore, agriculture is one of the leading global causes for climate change [6]. Assuming a balance between food production and consumption, global demand for agricultural land is a function of population size, production methods and waste, and dietary composition [7]. Although population characteristics like economic status, culture and religion largely influence consumption patterns [5,8–10], consumer choices can also strongly influence the environmental impact of food production [3]. Yet, accessible information on the environmental impact of dietary choices to guide individual consumers in their purchasing decisions is limited [3].

Many authors have attempted to quantify the influence of diet on land use and its impact on the environment, for instance by studying the land requirements and greenhouse gas emissions of different diets [6,11,12] or by performing life cycle analyses or land requirement assessments for specific food items [13–15]. However, none of these attempts provide information at a level that is accessible to individual consumers. Critically, we lack of an easily accessible and approachable

method for individual consumers to quantify the land use impact of their personal diet. This research attempts to bridge the gap between the scientific approach to quantifying land use and individual consumers. Alexander et al. have developed the Human Appropriation of Land for Food (HALF) index, which expresses the land area required for the global population to consume a specific diet as a percentage of the world land surface [16]. Although this index provides a relatively easy to understand metric of the land requirements associated with a certain diet, the authors use this index to quantify the land requirements of national diets. As individuals' diets might differ greatly from a certain national average, the current approach to calculating the HALF index cannot be used by individual consumers to assess the impact of their dietary choices.

The aim of this study is to develop an easy-to-use self-assessment tool that allows individual consumers to quantify the HALF index of their personal diet. In order to achieve this aim, our first objective is to develop a limited set of multiple-choice questions that represent those dietary parameters to which the original HALF index is most sensitive. Our second objective is to develop a parametrization of the original HALF index calculation to accept input from the set of multiple-choice questions. Our third objective is to validate the question-based HALF index against the original HALF index for a wide range of diets. In this regard, the proposed method resembles the user-friendliness of environmental footprint calculators like the WWF's carbon footprint calculator [17] or the BBC climate change food calculator [18], yet distinguishes itself from them in its sole focus on diet and its expression of the results in terms of the intuitive HALF index.

## 2. Materials and Methods

### 2.1. Question-Based HALF Index

The question-based HALF index calculator allows for the calculation of the HALF index based on a set of ten multiple-choice questions. The calculator was developed as a graphical user interface using the Tkinter package in programming language Python. The full source code can be found as Supplementary Material S2. Calculations underlying the question-based HALF index were based on the original HALF index, as developed by Alexander et al. [16]:

$$HALF = \frac{p_g * a_{app}}{a_{tls}} * 100, \tag{1}$$

where $p_g$ is the global population, $a_{app}$ is the agricultural area per person required to produce their food (in ha) and $a_{tls}$ is the total land surface area (in ha). The historic global population $p_g$ is based on counts from the UN World Population Prospects. In our approach we used the years 1850, 1975 and 2018 with population sizes of 1262 million, 4079 million and 7631 million, respectively [19]. The global ice-free land surface area $a_{tls}$ was taken to be 13,009 Mha [20]. As the factors $p_g$ and $a_{tiss}$ are not influenced by individual consumers, our set of questions solely focused on quantification of the factor $a_{app}$ according to

$$a_{app} = \left( \frac{p_{pb}}{yield_{plant}} + pasture + fish_{lu} + \frac{staple}{yield_{staple}} \right) * q, \tag{2}$$

where $p_{pb}$ is the mass of plant-based food product consumption per person without additional staple foods (kg/year), $yield_{plant}$ is the average crop yield (kg/ha/year), *pasture* is the pasture area required for consumption of ruminant animal products per person (ha), $fish_{lu}$ is the land use required for fish production per person (ha), *staple* is the mass of additional staple food consumption per person (kg/year), $yield_{staple}$ is the yield for the additional staple food (kg/ha/year), and $q$ is a dimension-less factor accounting for consumption quantity.

As the calculation of the HALF index is most sensitive to the consumption of animal products, our developed set of questions focused on this factor. The consumer is asked to provide information on how often they eat a portion of beef, pork, poultry, lamb or mutton, fish, cheese, milk and eggs in a week. The portion sizes of beef, pork, poultry, lamb or mutton, and fish were set at 100 g, representing

an average portion used in a meal; the portion size of cheese is 20 g, representing one pre-cut slice of cheese; the consumption of milk and other liquid dairy products is divided into portions of 250 g, corresponding to a glass of milk or a bowl of yoghurt; and the consumption of eggs is simply counted in whole eggs, assumed to be 50 g. The value assigned to the frequency is then multiplied with the portion size in grams to provide the number of grams consumed per food product per person per week.

In addition to the detailed questions on the consumption of animal products, two reference diets were developed to reflect dietary composition of plant-based products for a meat-based diet and a vegetarian diet. The dietary information was adapted from Pimentel and Pimentel for a meat-based diet and a vegetarian diet, based on an average American diet [21]. Here it was assumed that the calories and nutrients missing in the vegetarian diet from the consumption of meat and fish were compensated by increased consumption of plant-based products. If a user of the calculator enters they eat no meat in any form, they are assigned to the vegetarian reference diet, and if they do eat meat, they are assigned to the meat-eating reference diet. These diets were normalized to a total caloric intake of 3533 kcal/day including consumption of animal products [21]. An overview of the reference diets for the plant-based food sources is given in Table A1. These reference diets were supplemented with the calories and nutrition from the consumption of animal products. Additional staple foods, either grains or root vegetables, were used to supplement the diet until the total number of calories is 3533 kcal.

The quantity of consumption, irrespective of diet, is quantified relative to other consumers, and to this aim the user is asked if they eat more than average, about average, or less than average. Hereto, a factor $q$ is assigned that is multiplied with the total agricultural land needed to provide their food. If the user indicates they eat more than average, a value of 1.3 is assigned; if they indicate they eat about average quantities, a value of 1.0; and if they indicate they eat less than average, a value of 0.7.

*2.2. Feed Conversion Ratios*

Each animal product except fish was assigned a feed conversion ratio (see Table A2), which is the amount of feed required to produce one unit of the product [16]. It is assumed that the different animals have the same feed composition, yet different feed conversion ratios. For the production of one unit of cheese we assumed that seven units of milk are required [13]. The feed conversion ratios were then multiplied with the mass per food source in grams to obtain the amount of plant-based feed required in grams per food source. This is summed over all food sources, converted to kilograms, and converted to annual time scale to grant the total mass in kilograms of feed required per year.

The consumption of plant-based products consists of the feed for animals and the plant-based food sources as part of the reference diet. The total reference plant-based food sources were converted from grams per day to kilograms per year and then added to the total feed requirements (in kg/year). This value is then divided by the average crop yield of a certain year (in kg/ha/year) to give the total land requirements in hectares per capita. The average crop yield for 1850 is based upon the average yield of wheat across Europe [22]. The crop yields for 1975 and 2018 were based upon the yearly yields of the eight food crops with the highest harvested area, namely wheat, rice, maize, soybeans, pulses, barley, sorghum and millet [23]. These yields were weighed for their harvested area in that year and averaged to give the average crop yields [24]. The final average crop yields for the different years analyzed can be found in Table A3.

Ruminants, namely cattle and sheep, require additional pasture area. The amount of pasture required is adapted from the review of Nijdam et al. [13]. As there are large differences in pasture requirements for beef production between different production systems, with an average of 14 m$^2$/kg for industrial systems, 82.5 m$^2$/kg for meadows and 335 m$^2$/kg for extensive pastoral systems, the production systems were weighed for their share in global production before being averaged: industrial systems make up 6% of beef production and meadows and extensive pastoral systems were assumed to make up the rest in equal parts [25]. The pasture area required per food source of ruminants can be found in Table A2. The pasture requirements per gram were multiplied with the mass per food source in grams and converted to annual time scale to give the total pasture area required per food

source in m$^2$, which is then summed over all food sources and converted to hectares to give the total pasture area required in ha per person.

The share of aquaculture in the catch and production of fish has only become substantial in the past two decades and the consumption of fish therefore does not contribute to the HALF indices of 1850 and 1975 [26]. To reflect that aquaculture made up approximately half of total fish production in 2018, the fish consumption is multiplied with a factor of 5. The land use from fish consumption in grams per capita per year is then calculated by multiplying the modified fish consumption in grams by the land requirements for fish in ha/g and converting it to annual time scale. The land requirement is adapted from Nijdam et al. to be 4 m$^2$/kg/year [13].

## 2.3. Staple Foods

The mass of the chosen additional staple foods per capita per year needed to supplement the diet to 3533 kcal/day was calculated as:

$$staple = \frac{3533 - ref_{ppb} - a_p}{staple_{kcal}} * \frac{365}{1000} , \qquad (3)$$

where *staple* is the mass of additional staple foods per year (kg/year), $ref_{ppb}$ is the caloric intake of plant-based food sources in reference diet (kcal/day), see Table A1, $a_p$ is the caloric intake of animal products (kcal/day) and $staple_{kcal}$ is the average calories per gram of chosen staple food (kcal/g). The consumer indicates which type of staple food they eat most often, namely grains (e.g., rice, maize, pasta, bread) or root vegetables (e.g., potatoes, cassava). This choice then determines the yield value and the caloric value of the staple food, which can be found in Table A4. The caloric intake of animal products per day is determined by multiplying the consumed mass of animal products per day with their caloric value, as given in Table A5. The mass of the additional staple food is then divided by its yield (kg/ha/year) to give the total land area required to produce the additional staple crops.

## 2.4. Proof of Concept on Typical Diets

After the questions were developed, the question-based HALF indices for the world and twenty-two separate countries were compared to the original calculations of the HALF index by Alexander et al. [16]. These countries were chosen to be the countries with the three lowest and highest three HALF indices as found by Alexander et al., supplemented to include five countries from each of the regions Asia, Africa, Americas, and Europe, as well as the countries Australia and New Zealand. The average national diets were identified by dividing the total food supply quantities of different animal-based food products per country per year by the country's population and a factor of 52, yielding the amount consumed in grams per week [27,28]. Similarly, the average global diet was defined by dividing the total global food supply quantities by the global population. This quantity was then divided by the portion sizes of the different food products to arrive at the number of portions consumed per product per person per week, which was used to fill out the question-based HALF index calculator. The consumption of different food products under the different diets can be found in Table S1. Each country's daily caloric supply was compared to the global caloric supply to determine if quantities consumed were higher, equal to, or lower than the average consumption quantities [29]. All diets include grains as the staple food. The HALF indices of the countries as calculated in this study were plotted against the HALF indices found by Alexander et al. [16], and a linear regression analysis was performed to evaluate the relationship between the results.

Furthermore, four fictional example diets were defined: a meat-eating diet, a "flexitarian" diet, a vegetarian diet, and a vegan diet. A meat-eating diet was assumed to include meat consumption multiple times a day, a flexitarian diet was characterized by a reduced consumption of animal products with respect to the meat-eating diet, a vegetarian diet was characterized by no consumption of meat or fish while still containing other animal products such as dairy or eggs, and a vegan diet consisted of

merely plant-based foods and contains no animal products at all. These diets were determined by their consumption of animal products, and the example breakdown of their consumption of animal products is given in Table A6. These dietary descriptions were used to illustratively fill out the question-based HALF index calculator. The fictional diets were assumed to have an average consumption quantity and have grains as the main staple food. By using the calculator reference diet, these diets were all automatically normalized to a total caloric intake of 3533 kcal.

## 3. Results

### 3.1. The Question-Based HALF Index Calculator

The question-based HALF index calculator takes the form of a graphical user interface, based upon the Tkinter package in Python. The calculator enables the user to find out their personal HALF index for 1850, 1975 and 2018. The source code can be run in any programme capable of running Python, such as Jupyter, Spyder or a terminal window, or can be accessed and run online [30]. The user is greeted by a window briefly explaining the HALF index and subsequently asked ten questions about their dietary habits. The ten questions and possible answers are presented in (Table 1). After answering the questions, the user is provided with their personal HALF index for the years 1850, 1975 and 2018, taking into account the change in population size.

**Table 1.** Overview of the questions asked in the question-based HALF index calculator.

| Question | Possible Answers |
|---|---|
| 1. How much do you eat compared to others? | - Less than others<br>- Equal to others<br>- More than others |
| 2. How often do you eat a 100 g portion of BEEF on average? Assume a slice of lunch meat is 25 g. | - Never<br>- Rarely: 1–2 times a week<br>- Sometimes: 3–4 times a week<br>- Often: 5–7 times a week<br>- More than once a day |
| 3. How often do you eat a 100 g portion of PORK on average? Assume a slice of lunch meat is 25 g. | - Never<br>- Rarely: 1–2 times a week<br>- Sometimes: 3–4 times a week<br>- Often: 5–7 times a week<br>- More than once a day |
| 4. How often do you eat a 100 g portion of CHICKEN or TURKEY on average? Assume a slice of lunch meat is 25g. | - Never<br>- Rarely: 1–2 times a week<br>- Sometimes: 3–4 times a week<br>- Often: 5–7 times a week<br>- More than once a day |
| 5. How often do you eat a 100 g portion of LAMB or MUTTON on average? | - Never<br>- Rarely: 1–2 times a week<br>- Sometimes: 3–4 times a week<br>- Often: 5–7 times a week<br>- More than once a day |
| 6. How often do you eat a 100 g portion of FISH on average? | - Never<br>- Rarely: 1–2 times a week<br>- Sometimes: 3–4 times a week<br>- Often: 5–7 times a week<br>- More than once a day |
| 7. How often do you eat a 20 g portion of CHEESE on average? Assume a pre-cut slice is 20 g. | - Never<br>- Rarely: 1–2 times a week<br>- Sometimes: 3–4 times a week<br>- Often: 5–7 times a week<br>- More than once a day |

**Table 1.** *Cont.*

| Question | Possible Answers |
|---|---|
| 8. How often do you consume a 250 g portion of cow MILK or other LIQUID DAIRY products (e.g., yoghurt) on average? | - None<br>- Less than once a day<br>- Once a day<br>- Twice a day<br>- More than twice a day |
| 9. How many EGGS do you eat per week on average? | - None<br>- A few: 1–2<br>- Several: 3–5<br>- Many: more than 5 |
| 10. Which staple food do you eat most often? | - Grains (e.g., rice, maize, pasta, bread)<br>- Root vegetables (e.g., potatoes, cassava) |

### 3.2. Proof of Concept

The results from the question-based HALF index calculator for the national and global HALF indices were plotted against the HALF indices found by Alexander et al. [16], as can be seen in Figure 1. A regression analysis showed a significant correlation ($p < 0.001$, $r^2 = 0.94$) between the HALF indices found in this study and the HALF indices found by Alexander et al., with this relationship described by the linear regression equation $y = 0.96x + 2.44$. The 95% confidence interval of the regression slope is 95% CI [0.85–1.09]. The original HALF index calculated for New Zealand was considered a mistake in the calculation of the original HALF index and therefore excluded in this analysis.

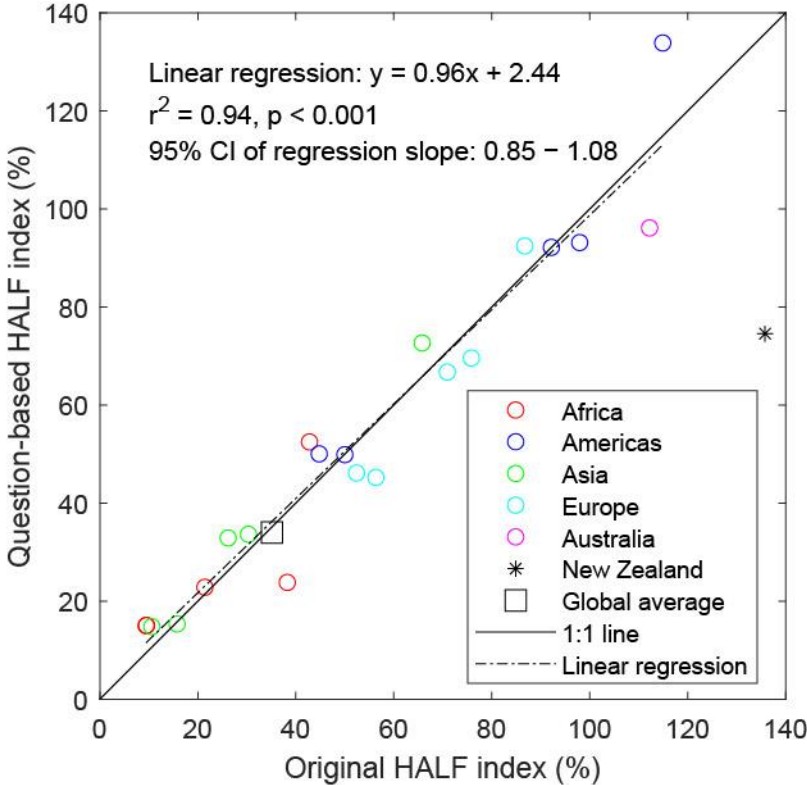

**Figure 1.** Plot of the HALF indices as calculated in this study versus the original HALF indices as calculated by Alexander et al. [16] for 23 different national diets and the global average diet. The solid line shows the 1:1 line, and the dashed line shows the linear regression line.

Question-based HALF indices were then calculated for the global average diet and for four fictional diets. The question-based HALF closely matched the observed current global land use for

agriculture and the original global HALF as calculated by Alexander et al. [16] (Figure 2a). Moreover, the question-based HALF index lowered with reduced consumption of animal-products (Figure 2b). The difference in question-based HALF indices between the vegetarian diet and the vegan diet was small compared to the difference between the diets containing meat (meat-eating diet and flexitarian diet) and the diets containing no meat (vegetarian diet and vegan diet). The question-based HALF indices of both a meat-eating diet and a vegan diet increased over time due to population growth (Figure 2c). For both diets the question-based HALF index approximately doubled between 1850 and 2018. Notably, the question-based HALF index for a vegan diet in the year 2018 is lower than that of a meat-eating diet in the year 1850.

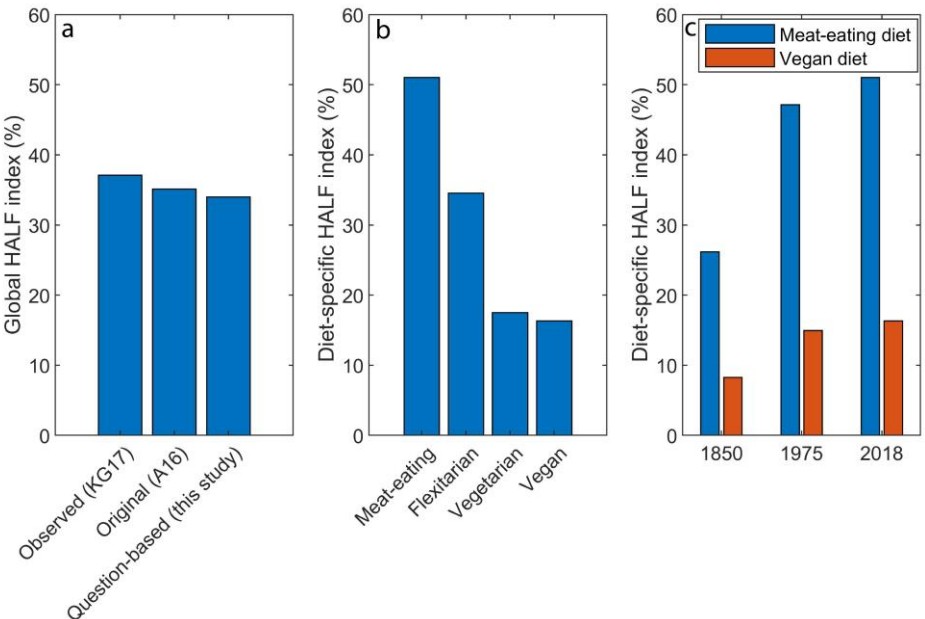

**Figure 2.** (**a**) The observed global HALF index in 2015 as calculated by Klein Goldewijk et al. (KG17), the global HALF index in 2016 as calculated by Alexander et al. (A16), and the global HALF index in 2018 as calculated in this study [1,16]; (**b**) The HALF indices for a meat-eating, flexitarian, vegetarian and vegan diet as calculated in this study; (**c**) The HALF indices for a meat-eating and a vegan diet overt time as calculated in this study.

## 4. Discussion

The question-based HALF index calculator enables a consumer to quantify the global land use impact of their personal diet in an easy-to-use, straightforward manner. Although the original HALF index [16] is attractive for its intuitive interpretation, the quantitative methodology required in the calculation is not easily reproduced by a typical consumer trying to quantify the impact of his or her personal diet. The question-based HALF index calculator instead permits the influence of individual dietary choices by combining the HALF framework with the user-friendliness of existing footprint calculators, such as the carbon footprint calculator developed by WWF, the ecological footprint calculator of the Global Footprint Network, or the GRACE Communications Foundation's water footprint calculator [17,31,32].

There is a strong significant positive correlation between the HALF indices calculated using the question-based HALF index calculator and the results found by Alexander et al. [16], which suggests that the calculator as developed in this study is a valid approach to determine HALF indices. As the 95% confidence interval of the regression slope includes the value 1.0, there is very little bias in the results. In our results, we observed the original HALF index for New Zealand to be anomalously high. For all products included in the question based HALF index calculator except for lamb/mutton, the consumption of New Zealand is more in line with that of The Netherlands or Norway than with

Australia or Argentina. To illustrate, the average beef consumption in Norway, New Zealand and Argentina is, respectively, four, four and ten portions a week [27]. Hence, the calculation of New Zealand in the original HALF index was considered an error and excluded from our regression analysis.

The results for the global question-based HALF index and the results for the global HALF index found by Alexander et al. [16] are both within an 8% difference with respect to the observed HALF value. The calculated HALF indices for the four fictional diets, namely a meat-eating, flexitarian, vegetarian and vegan diet, indicate that diets with a higher consumption of animal products have higher agricultural land requirements than diets with a reduced consumption of animal products. This adheres to the trend found across the literature, for example by Alexander et al., Peters et al., Stehfest et al., and Tilman and Clark [6,11,16,33]. Reducing the consumption of animal products and therefore moving from a meat-eating diet to a flexitarian diet could mean a reduction in HALF index by a third, while switching to a vegetarian or vegan diet could mean reducing the HALF index by two thirds, which quantitatively matches the results found by Poore and Nemecek [3].

The results as discussed above are based on certain assumptions with associated uncertainties in the analysis. As the average crop yields for 1975 and 2018 are based on only eight crops, they merely approximate the true average crop yield. Still, these crops together make up two thirds of the total harvested area and the estimates can therefore be assumed to be meaningful [23]. The average crop yield for 1850 was based solely on wheat yields in Europe because wheat has consistently been the crop with the highest harvested area [24]. The limited scope is due to the lack of available historic data on crop yields for different crops or regions. Due to the commencement of the Industrial Revolution in Europe in the nineteenth century, the continent might have had higher crop yields than other regions at the time. Although the global land surface area includes land unsuitable for agriculture, it was preferred to use this value instead of only the area of "suitable land" due to differences in opinion and measurement on what constitutes such suitable land [16]. This does mean that a HALF index of close to a hundred reflects a highly unrealistic value. Furthermore, the question-based HALF index calculator does not take into account differences in production systems. However, while this could make a difference for an individual's land use or emissions, the global character of the analysis conceptually supports using average properties of production systems. Conveniently, this allows quantification of the impact of diet without considering in detail the complicating factor of differences in production systems [16]. Still, the global nature of the calculator might lead to a misrepresentation of a consumer who consumes mainly local produce due to potential differences in efficiency between local and global average production systems.

Our study did not consider food waste by consumers, as this quantity is notably lower than the available calories per person [34]. An individual's HALF index as calculated by the calculator therefore does not include their personal food waste. The national diets, however, are based on the food supply per capita as there is no record of the true personal consumption excluding food waste. Consequently, the HALF index for national diets implicitly includes the food waste at the household level, and the true HALF index based on consumption excluding food waste might be 9–12% lower in the US and the Netherlands and 1–3% lower in India [35]. The national diets are solely characterized by the consumption of animal products, while the consumption of plant-based products is assumed to be that of the reference diet with a total caloric intake of 3533 kcal/cap/day. This therefore might not perfectly reflect the true national consumption patterns, as the reference diet is based on an average American diet as defined by Pimentel and Pimentel [21]. It was assumed that a lower consumption of animal products was compensated by solely a higher staple intake in order to maintain a total caloric intake, which might have underestimated the HALF index as grains and root vegetables are less land intensive than other food groups as fruits or vegetables [33].

Despite these considerations, the proof of concept shows that the HALF indices for a meat-eating and a vegan diet have doubled since 1850. Based on population growth alone, a six-fold increase would be expected, yet the yield improvements have balanced out part of this growth. This agrees with Kastner et al., who found that technological advancements have only partly balanced out the cropland

increases due to population growth and dietary change [36]. Arguably, the influence of dietary choices has the chance to offset the influence of historic population growth as the question-based HALF index for a vegan diet for the year 2018 is lower than the question-based HALF index of a meat-eating diet for the year 1850. Hence, dietary change has the potential to reduce agricultural land use, if the right choices are made.

The question-based HALF index calculator can be used to increase awareness and act as a learning opportunity about the impact of dietary choices on land use. Other footprint calculators have already been introduced as a tool to create public awareness and have been used as a learning and teaching tool in sustainability education [37,38]. The increased understanding of the impact of diet that the question-based HALF index calculator could help cultivate, can shape the foundations for widespread dietary change towards reduced consumption of animal products. The question-based HALF index calculator could help reduce two of the barriers to reduced meat consumption as identified by Macdiarmid et al. [39], namely, a lack of awareness of the association between meat consumption and climate change, and the perceptions of personal meat consumption playing a minimal role. The main implication of this research is that the question-based HALF index calculator provides a valid extension of the original HALF index and allows individual consumers to assess the land use impact of their personal diet. On the basis of ten simple questions, the calculator is able to determine the HALF index accurately, thereby quantifying the complex interactions between diet and agricultural land use in an easy-to-use way. Future research could apply a similar approach to allow individuals to investigate the link between their diet and other aspects of their environmental footprint, such as $CO_2$, water use, or energy input. Further research could also be done on the effectiveness of such environmental calculators in changing the user's behavior or awareness.

**Supplementary Materials:** The following are available online at http://www.mdpi.com/2071-1050/12/24/10597/s1. Table S1: Number of portions consumed weekly of different food products in different countries, Source code S2: Source code of the question based HALF index calculator, which is available online at http://www.mdpi.com/2071-1050/12/24/10597/s2.

**Author Contributions:** Conceptualization, methodology, validation, formal analysis, investigation, resources, data curation, visualization and project administration, M.H. and H.J.d.B.; Software and writing—original draft preparation, M.H.; Writing—review and editing, and supervision, H.J.d.B. All authors have read and agreed to the published version of the manuscript.

**Funding:** This research received no external funding.

**Conflicts of Interest:** The authors declare no conflict of interest.

## Appendix A

Table A1. Reference diet of plant-based food sources for a meat-eating and a vegetarian/vegan diet, adapted from Pimentel and Pimentel [21].

| | Meat-Eating Diet | | Vegetarian/Vegan Diet | |
|---|---|---|---|---|
| | **Consumed Per Day (g/day)** | **Energy (kcal/day)** | **Consumed Per Day (g/day)** | **Energy (kcal/day)** |
| Food grain | 312.33 | 849 | 416.44 | 1132 |
| Pulses | 11.78 | 40 | 20.55 | 70 |
| Vegetables | 654.79 | 147 | 783.56 | 155 |
| Oil crops and vegetable oils | 82.19 | 619 | 90.41 | 665 |
| Fruits | 298.63 | 122 | 306.85 | 122 |
| Nuts | 8.49 | 23 | 10.96 | 30 |
| Sugar and sweeteners | 202.74 | 686 | 202.74 | 30 |
| Total plant-based food sources | 1570.96 | 2486 | 1831.51 | 2860 |

**Table A2.** Feed conversion ratios and pasture area requirements for different animal-based food sources [13,16].

| Food Source | Feed Conversion Ratio | Pasture Area Required (m²/g) |
|---|---|---|
| Beef | 25 | 0.197 |
| Pork | 6.4 | 0 |
| Poultry | 3.3 | 0 |
| Lamb | 15 | 0.024 |
| Eggs | 2.3 | 0 |
| Milk | 0.7 | 0.001 |
| Cheese | 4.9 | 0.007 |

**Table A3.** Average crop yield in kg/ha/year for different years [22–24].

| Year | Average Crop Yield (kg/ha/year) |
|---|---|
| 1850 | 780 |
| 1975 | 1590 |
| 2018 | 3530 |

**Table A4.** Yearly yields (kg/ha/year) and caloric values (kcal/g) of staple foods [21,24,40,41].

| | Grains | Root Vegetables |
|---|---|---|
| 1850 yield | 780 | 10,500 |
| 1975 yield | 1700 | 11,300 |
| 2018 yield | 3700 | 13,400 |
| Caloric value | 2.72 | 1.12 |

**Table A5.** Calories per gram of different animal-based food sources [41].

| Food Source | Calories Per Gram |
|---|---|
| Beef | 1.61 |
| Pork | 3.79 |
| Poultry (chicken) | 1.83 |
| Lamb | 1.58 |
| Egg | 1.32 |
| Milk (semi-skimmed) | 0.452 |
| Cheese | 3.7 |
| Fish | 2.42 |
| Roots and tubers | 1.12 |

**Table A6.** Animal-product consumption under different fictional example diets.

| | Meat-Eating | Flexitarian | Vegetarian | Vegan |
|---|---|---|---|---|
| Portions of beef (100 g) | 3 | 1 | 0 | 0 |
| Portions of pork (100 g) | 4 | 1 | 0 | 0 |
| Portions of poultry (100 g) | 5 | 1 | 0 | 0 |
| Portions of lamb and mutton (100 g) | 0 | 0 | 0 | 0 |
| Portions of fish (100 g) | 2 | 1 | 0 | 0 |
| Portions of cheese (20 g) | 7 | 7 | 7 | 0 |
| Portions of milk and liquid dairy (250 g) | 14 | 7 | 7 | 0 |
| Number of eggs | 7 | 4 | 4 | 0 |

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
