# Peer review of "A Question-Based Method to Calculate the Human Appropriation of Land for Food (HALF) Index"

_sustainability, doi:10.3390/su122410597_

Round 1

Reviewer 1 Report

The paper: “A question-based method to calculate the Human Appropriation of Land for Food (HALF) index” is complete in all its scientific elements. All the parts are relevant to the topic of the research. The methodological approach used and the results are clear. Furthermore, the methodology adopted could be easily emulated for further researches in this field. The idea of the paper is interesting and I think that it can be improved, the authors should better highlight the limitations of the study and possible future researches on the topic, as well.

My overall evaluation of the paper is positive; however, I think the paper needs some small revisions:

The section introduction must be reorganized. Authors should better clearly identify the research gap, the research question and the research objective of their study.

Finally, authors should write a section of the conclusions emphasizing the main implications of the study, the limitations and future research.

Reviewer 2 Report

For who this research is important and useful? Who, as an individual consumer will calculate  his own consumption pattern with the use of proposed method.

Reviewer 3 Report

The paper presents a very interesting idea of calculating the impact of diet on the global ecosystem by the potential change in the area of agricultural land needed for providing food.

Even the overall idea behind the paper seems to be very interesting for readers authors should provide limitation of the applied methodology.

Even the concept of calculation HALF based on the questionnaire seems to be very reasonable and justified for analysis at globe scale it might be misleading when analysing results at country levels. Authors for the simplicity reasons decided to take the average characteristics of food production technology, i.e. yields, feed conversion ratio etc. In fact, there are huge differences between countries in the technology of agricultural production. Technologies used in agricultural production differs due to climatic conditions, overall development level of the country, represented values (e.g. promoting/subsidizing organic or precision farming), the structure of farming sector (small farms/ big farms, privately/state-owned) etc.

Those technological differences result in different yields, e.g. 3,7t/ha for grains seems to be a very low yield for Western European Countries, different composition of the animal production system, different feed conversion ratio.

 The same applies, what authors already mentioned to average diets, which, e.g. requires energy intake, could differ across the countries.

As including all those differences would be very demanding, I would suggest adding short paragraph describing limitations of the applied methodology.
